# Streamflow Measurement Using Mean Surface Velocity

**Yen-Chang Chen [1], Yung-Chia Hsu [2],\* and Eben Oktavianus Zai [1]**

[1] Department of Civil Engineering, National Taipei University of Technology, Taipei 10617, Taiwan;
yenchen@ntut.edu.tw (Y.-C.C.); d91521012@g.ntu.edu.tw (E.O.Z.)

[2] Department of Civil Engineering, National Central University, Taoyuan 32001, Taiwan

\* Correspondence: ychsu1978@g.ncu.edu.tw; Tel.: +886-90-666-18-39

**Abstract:** This study developed an efficient discharge measurement method that can be applied to estimate the streamflow of natural streams and artificial channels. The conventional methods that apply current meters to measure discharge are costly, time-consuming, and labor-intensive. Owing to a shortage of observers in streamflow measurement and for the safety of hydrologists and with advances in measurement techniques, many have strongly suggested the use of non-contact methods when determining streamflow. The non-contact methods that use floats or surface velocity radar to determine the streamflow are becoming more and more popular especially during periods of high water. However, it is not easy to estimate the surface velocity coefficient of each vertical directly for determining the mean velocity in each subsection. As the relationship between the mean surface velocity and mean velocity of a stream cross-section is constant, an efficient and accurate non-contact method of streamflow measurement could be further developed. Thus, streamflow can be estimated by the constant, the mean surface velocity, and cross-sectional area of a stream. The mean velocity of a cross-section, used for parameter calibration, is usually obtained from the discharge made based on the velocity-area principle and cross-sectional area. The surface velocity was measured at the vertical that is then used to estimate mean velocity of a subsection. Once the parameter is determined, streamflow can be obtained from the surface velocity. This approach was further applied to a natural stream and an artificial channel. Measurements were made to verify the reliability and accuracy of the proposed approach. The results show that the relationship between mean channel velocity and mean surface velocity is very stable in both a natural stream and an artificial channel because the streamflow differences, given by the proposed and the conventional method, are relatively insignificant. As a result, mean surface velocity can be used to determine the streamflow quickly and provides for a reliable and accurate measurement of streamflow.

**Keywords:** discharge measurement; mean surface velocity; non-contact measurement; acoustic doppler flowmeter; magnetic-inductive current meter

## 1. Introduction

Conventional methods of river discharge measurement apply the velocity-area principle [1] for field measurements while the mid-section method [2] is used to calculate the streamflow. The conventional method involves dividing a river cross-section into several subsections. In each subsection, the mean velocity and water depth are measured along the vertical to obtain the discharge of the subsection. The streamflow is the sum of the discharge measurements of all subsections. The conventional method is usually a contact method, requiring a current meter, a sounding weight, and hydrologists on site making this method costly, time-consuming, and labor-intensive. This method is not suitable for tidal streams and during the high water.

Many instruments have been invented with the goal of improving conventional methods, allowing for the rapid and accurate measurement of flow velocity and water depth. An acoustic Doppler current profiler (ADCP) which applies the Doppler effect is a

relatively new instrument and has been widely considered as a method used to replace mechanical current meters for velocity measurement. Simpson [3] used a broad-band ADCP that is much faster for accurately measuring tidally affected flow than conventional methods. Boiten [4] applied ADCPs to measure discharges in open channels, according to the velocity-area principle. Costa et al. [5,6] used a boat-mounted ADCP to measure discharge for converting surface-velocity to mean velocity. Muste et al. [7] analyzed velocity profiles collected by ADCPs to propose complementary software for better support of hydraulic investigation requirements. Chauhan et al. [8] showed that the relative error of the discharge measurement is very small with the ADCP compared to the conventional method. Oberg and Mueller [9] showed that ADCP streamflow measurements are unbiased when compared to the discharges obtained by current meter, stable rating curves, salt-dilution, and acoustic velocity meter. Flener et al. [10] used the multidimensional spatial flow patterns measured by an ADCP installed on a remotely-controlled boat to monitor a spring flood. Ground penetration radar [11,12], lidar [13], pressure sensors, and sonar systems [14,15] have been developed to replace the conventional sounding weights during water depth measurement. Although these modern instruments are costlier, they can be applied to provide data when conventional instruments cannot, with the extra benefits of reducing the overall cost and time required [10,16].

It is better to not put hydrologists and equipment in contact with the water, given concerns for personal safety and efficiency. The US Geological Survey [16] has also suggested that in the future gaging stations can use remote sensing outside the flow of water to measure water stages, cross-section, and velocity, and so on. In the past and currently, floats are most often used for surface velocity measurements. The principal sources of error inherent in determining surface velocity in this way are wind and flow conditions. With the recent development of new technology, non-contact instruments [5,6] used for measuring surface flow velocity and to estimate discharge have gradually been developed, among which, surface velocity radar (SVR, [17–20]) and particle image velocimetry (PIV, [21]) are the main ones. The most important issue when applying surface velocity to estimate discharge is the choice of a surface-velocity coefficient. In a natural channel a surface-velocity coefficient of 0.85 or 0.86 is typically used to compute mean flow velocity [2]. However, the surface-velocity coefficient varies with the positions of the verticals from the river bank. Generally, the closer to the bank, the larger the surface-velocity coefficient. In addition, with higher water flow the maximum velocity will occur below the water surface, so the surface-velocity coefficient will also become larger. The velocity distributions can be used to determine the surface-velocity coefficient. The logarithmic distribution [22,23] and probabilistic velocity distribution [24] are commonly used to determine the surface-velocity coefficient. The mean velocity of the vertical can be estimated by the surface-velocity coefficient and surface velocity, and then the velocity-area principle can be used to estimate streamflow.

Although using surface velocity to estimate river discharge is quite efficient, it is still necessary to select surface-velocity coefficients for all verticals to accurately estimate the streamflow; however, it is always a difficult task to determine the best coefficients. Therefore, this study proposes a method which uses the average surface velocity and only one surface velocity coefficient to efficiently and accurately estimate streamflow.

## 2. Materials and Methods

### 2.1. Relation between Mean Velocity and Mean Surface Velocity

The mid-section method is often used to calculate streamflow. Figure 1 illustrates that in the mid-section method the mean velocity on the vertical represents the mean velocity in a subsection. The subsection area extends laterally from half the distances from the preceding vertical to half the distance to the next as shown by the hatched area in Figure 1. $b_{n-1}$ in the Figure 1 is the distance from initial point to the n-1$^{th}$ vertical; $d_{n-1}$ is the water depth at vertical n-1; $\overline{u}_{n-1}$ is the mean velocity of the n-1$^{th}$ vertical. Therefore, each

subsection is rectangular. The subsectional discharge $q_i$ and the subsection area $a_i$ were calculated using (1) and (2), respectively:

$$q_i = \overline{u}_i \left( \frac{b_{i-1} + b_{i+1}}{2} \right) d_i \tag{1}$$

$$a_i = \left( \frac{b_{i-1} + b_{i+1}}{2} \right) d_i \tag{2}$$

where $b_i$ is the distance from the initial point to vertical $i$; $d_i$ is the depth of flow at vertical $i$; and $\overline{u}_i$ is mean velocity at vertical $i$. The observed discharge ($Q_{obs}$) and cross-sectional area ($A_{obs}$) can be represented as (3) and (4), respectively:

$$Q_{obs} = \sum_{i=1}^{n} q_i \tag{3}$$

$$A_{obs} = \sum_{i=1}^{n} a_i \tag{4}$$

Thus, the observed mean velocity ($\overline{u}_{obs}$) was calculated using (5):

$$\overline{u}_{obs} = \frac{Q_{obs}}{A_{obs}} \tag{5}$$

The subsectional discharge can be obtained using (6):

$$q_{si} = u_{si} \left( \frac{b_{i-1} + b_{i+1}}{2} \right) d_i \tag{6}$$

where $u_{si}$ is the surface velocity on vertical $i$. If the cross-section of a stream is a rectangle or close to a rectangle, and the intervals between the verticals are equal; then the area of each subsection will be equal as shown in (7):

$$q_{si} = u_{si} a \tag{7}$$

where $a$ is the area of a subsection when the width and depth of the subsection are the same. The subsection discharge is given by (8):

$$Q_s = \sum u_{si} a = a \sum u_{si} = an\overline{u}_s = \overline{u}_s A_{obs} \tag{8}$$

where $Q_s$ is discharge estimated by surface velocity; and $\overline{u}_s$ is the mean surface velocity. However, (8) is not valid for estimating stream discharge. A surface-velocity coefficient must be applied to (8) to relate the data to the actual discharge amount as shown in (9):

$$Q_{obs} = \alpha Q_s \tag{9}$$

where $\alpha$ is surface-velocity coefficient. Thus:

$$\alpha = \frac{Q_{obs}}{Q_s} = \frac{\overline{u} A_{obs}}{\overline{u}_s A_{obs}} = \frac{\overline{u}}{\overline{u}_s} \tag{10}$$

(10) reveals that the relationship of mean cross-sectional velocity and mean surface velocity is a straight line going through the origin.

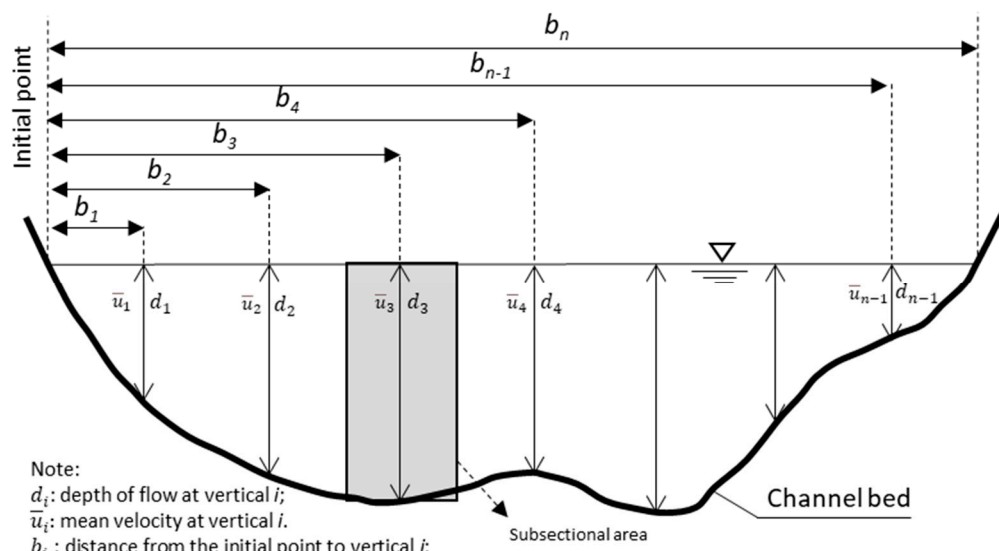

**Figure 1.** Mid-section method of computing cross-section area and stream discharge.

### 2.2. Estimation of Surface Velocity with Velocity Distribution Based on Probability

The surface velocity can be measured directly by SVR in most conditions. However, when the channel width is not enough for accommodating the instrument, one can measure the velocity profile on each vertical and use the velocity distribution equation to estimate the surface velocity. This study applied the probabilistic velocity distribution to estimate the surface velocity [24], which is shown in (11):

$$\frac{u}{u_{max}} = \frac{1}{M} ln \left[ l + (e^M - 1) \frac{\xi - \xi_0}{\xi_{max} - \xi_0} \right] \tag{11}$$

where $u_{max}$ is the max velocity; $M$ is a parameter; $\xi$ is the isovel in Figure 2 [25]; $u$ is the velocity at $\xi$; $\xi_{max}$ and $\xi_0$ are the values of $\xi$ at which $u=u_{max}$ and $u=0$, respectively. In addition, a $\eta - \xi$ coordinate system can be used to describe the velocity field with a set of isovels, in which $\xi$ and $u$ has a one-to-one relationship, meaning that the velocities are the same on $\xi$, unlike the Cartesian coordinate system where the same velocity values can occur in difference locations. The $\xi$ on the vertical line is shown in (12):

$$\xi = \frac{y}{D - h} exp \left( 1 - \frac{y}{D - h} \right) \tag{12}$$

where y is the vertical distance from the channel bottom; $D$ is the water depth; and $h$ indicates the location where the max velocity occurs. When $h \leq 0$, the max velocity occurs on the surface; when $h \geq 0$, the max velocity occurs below water surface $h$. Using the velocity profile and (11), nonlinear regression can be employed to estimate the parameter in (11) and the surface velocity for each vertical.

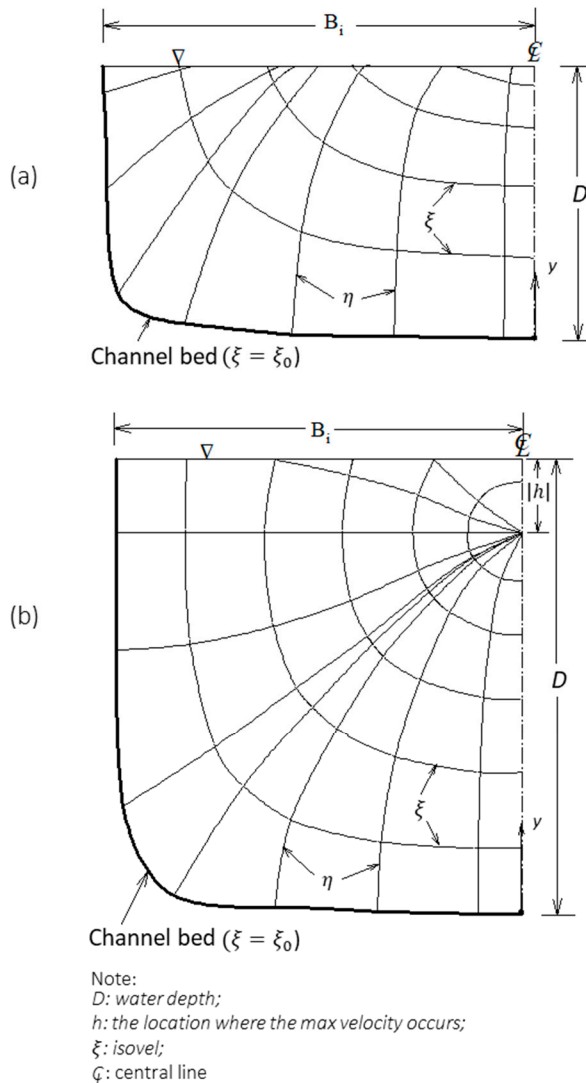

**Figure 2.** Velocity field in the $\eta - \xi$ coordinate system with a set of isovels. (**a**) $h \leq 0$; (**b**) $h > 0$.

*2.3. Estimation of Cross Section Area and Discharge*

In practice, there are many approaches for cross-sectional area estimation under different conditions. For a fixed artificial channel, the water stage is consistent enough that it can be used for estimating the cross-sectional area. For a stable channel bed without obvious erosion and sediment deposition, the relationship between the water stage and the cross-sectional area can be used to estimate the cross-sectional area of the river. For an unstable channel bed, the cross-sectional area can be estimated from the water depth of the verticals [26] using (13):

$$A_{est} = b(d - c)^e \tag{13}$$

where $A_{est}$ is the cross-sectional area estimated by water depth; $d$ is the water depth of a vertical; and $b$, $c$, and $e$ are coefficients.

Once $\alpha$ and $A_{est}$ are obtained, then the streamflow can be promptly evaluated from the surface velocities of the verticals.

$$Q_{est} = \alpha \overline{u}_s A_{est} \tag{14}$$

where $Q_{est}$ is the streamflow estimated by mean surface velocity.

The proposed approach only requires a SVR to obtain the mean surface velocity for the discharge ($Q_{est}$) measurement. If the bed does not change too much the cross-sectional

area ($A_{est}$) estimated by water depth may remain feasible. If not, one may also need an efficient and non-contact method, such as the GPR method [12], for fast cross-sectional measurement to update the $A_{est}$ function. As for the conventional approach (mid-section method), it requires obtaining the area and mean velocity of each subsection in order to obtain the subsectional discharge ($q_i$) and the observational discharge $Q_{obs}$.

### 3. Case Study: Study Sites and Data Collection Methods

This study applied field data collected in a natural river and an artificial channel to verify the proposed approach. The study sites are located along Nankang River at Guanyin Bridge in central Taiwan and along Longen Channel in Hsinchu, northern Taiwan (Figure 3).

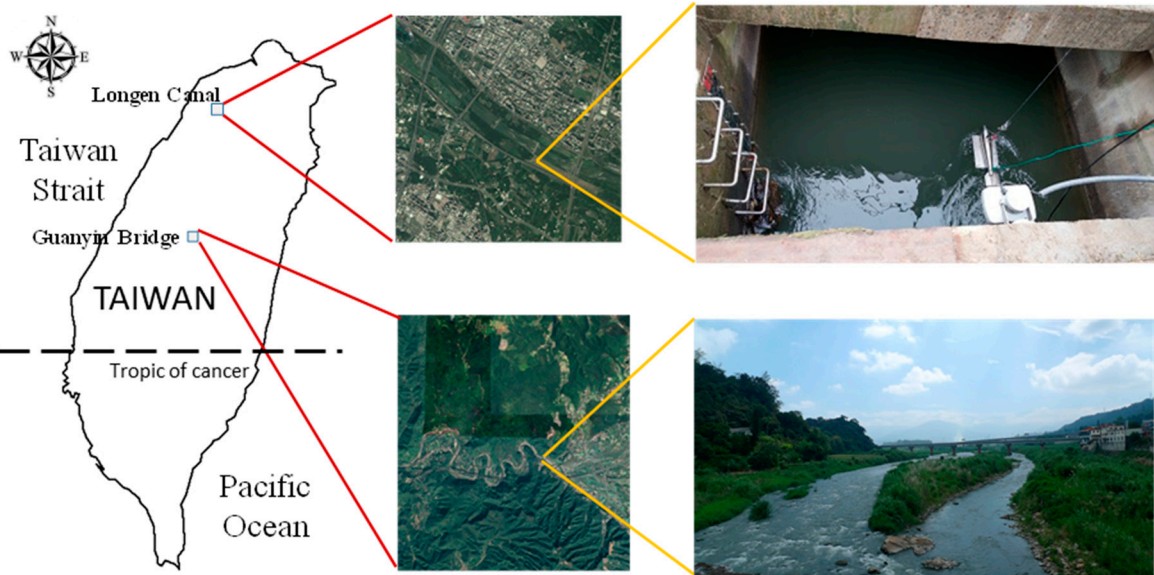

**Figure 3.** Study sites for discharge measurement.

The Nangang Creek serves as the main tributary of the Wu River, with a length of about 37.4 km and a catchment area of about 438.14 km². It originates from the western foothills of the Hehuan Mountain at an elevation of 3417 m. Wu River is one of the most important rivers in Taiwan, providing vast amounts of water for industrial, agricultural and domestic uses. Thus, the Third River Management Office set up a gauge station at the Guanyin Bridge for the purposes of water resources and flood management.

The Nangang Creek at the Guanyin Bridge, where the measurements were taken, is located near the geographical center of Taiwan, with water flowing from east to west. The channel of Nangang Creek near the Guanyin Bridge is divided into left and right channels. As shown in Figure 4, both the two channels are considered rectangular in cross-section, particularly during the periods of high water, while the left channel is narrower in width and shallower in depth. The collision of tectonic plates has uplifted the terrain of Taiwan over time, while floods brought by typhoons always wash the riverbeds, causing riverbeds to become unstable and change shape frequently. An electromagnetic current meter was used to estimate the water discharge at the Guanyin Bridge. The distance between two successive verticals was 3 m. The velocity observations were made based on the water depth of each vertical. When the water depth was greater than 0.6 m, the two-point method was used, while when the water depth was less than 0.6 m, the six-tenths depth method was used. This study also used a vehicle-mounted SVR unit to measure the surface velocity at each vertical (Figure 5a). A total 23 discharge measurements were made in 2018.

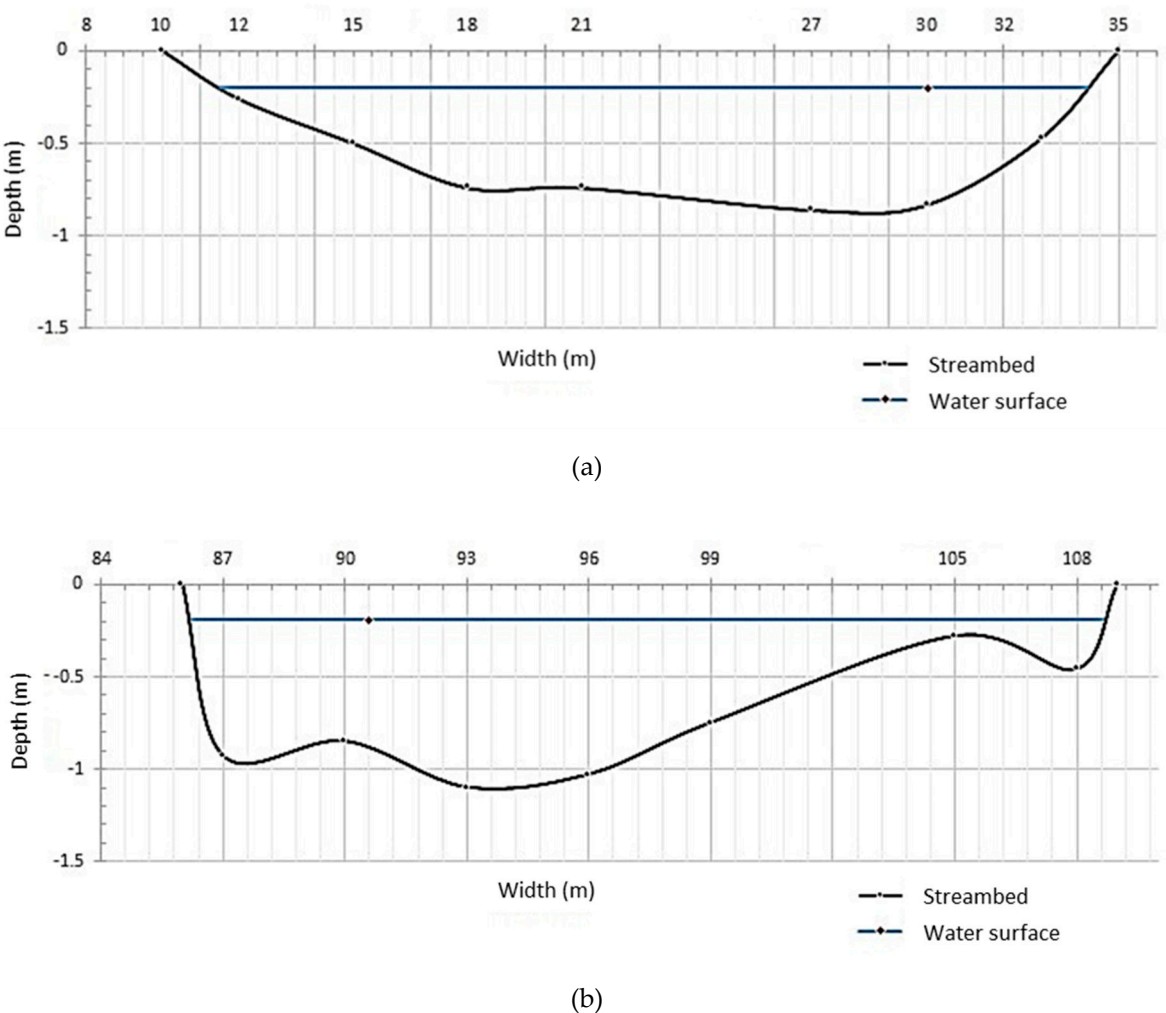

(a)

(b)

**Figure 4.** Channel cross sections of the Nankang River at the Guanyin Bridge; (**a**) left channel; (**b**) right channel.

The water in Longen Channel comes from Longen Weir in the middle reach of the Touqian River. Its main function is to divert water from the Touqian River for irrigation and domestic use purposes. The measurement location in Longen Channel was about 1 km downstream from the water intake location. The channel at this location was formed by a box culvert with no place for mounting instruments. Therefore, the top of the box culvert was opened about 1 m wide to allow for discharge measurements. The cross-section of Lungen Channel is rectangular with a width of 2.6 m. Due to the narrow width, the SVR unit can be easily affected by the side walls. With slow water flow, the water surface is very calm, which also makes the SVR unit unable to measure the surface velocity. Therefore, an Argonaut SW acoustic Doppler Flowmeter (Sontek, San Diego, CA, USA) was used to measure the velocity profile, and then a probabilistic velocity distribution equation was used to estimate the surface velocity. The SW Flowmeter was a pulsed Doppler Current profiling system designed for measuring water velocity profiles using three acoustic beams (Figure 5b). The slanted beams, Beams 1 and 2, measure the water velocity in two dimensions, and the down-looking beam measures water depth. An SW Flowmeter is usually mounted in a channel bottom. In this study, the SW Flowmeter was installed under the sounding weight, so that the velocity profile was measured from top to bottom. The velocity observation locations in Longen Channel are shown in Figure 6. The velocity distribution was measured on eight vertical lines, the distance between the verticals was 0.3 m, and the velocity was measured at ten points on each vertical. Table 1

shows the discharge and water depth measurements of eight runs during different water stages having covered the upper and lower water supply capability of Longen Channel. Based on Figure 6, the velocity measurements of Run 4 in Table 1 are shown in Figure 7.

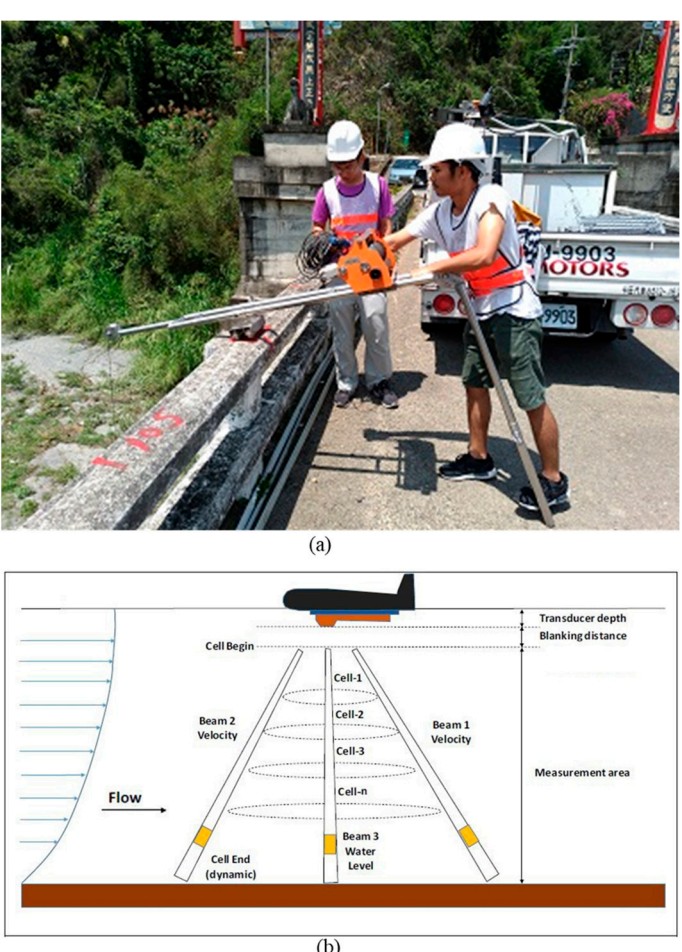

(a)

(b)

**Figure 5.** Instruments for measuring discharge. (**a**) A magnetic-inductive current meter and a SVR are used at the Guanyin Bridge; (**b**) A down-looking SW integrated with a sounding weight is used to measure the velocity profiles in the Longen Channel.

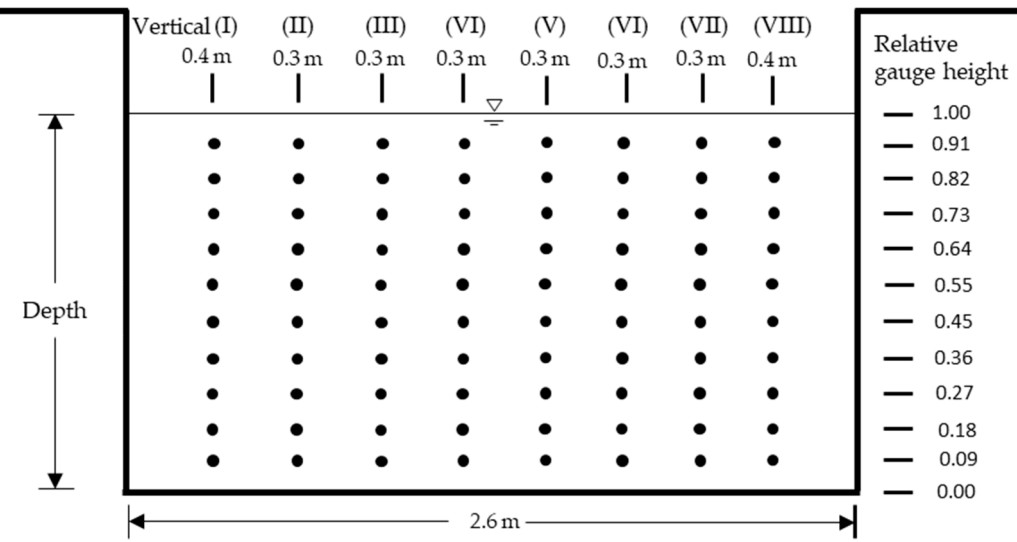

**Figure 6.** Discharge measurement at the Longen Channel.

**Table 1.** Discharge measurement of the Longen Channel.

| Run | Date | Depth (m) | Q_obs (Observed Discharge m³/s) |
|---|---|---|---|
| 1 | 22 February 2019 | 0.87 | 0.43 |
| 2 | 6 March 2019 | 0.98 | 0.92 |
| 3 | 8 March 2019 | 1.47 | 3.80 |
| 4 | 14 March 2019 | 1.43 | 4.24 |
| 5 | 20 March 2019 | 1.15 | 2.04 |
| 6 | 25 March 2019 | 1.31 | 3.16 |
| 7 | 3 May 2019 | 1.52 | 3.79 |
| 8 | 23 May 2019 | 1.48 | 4.35 |

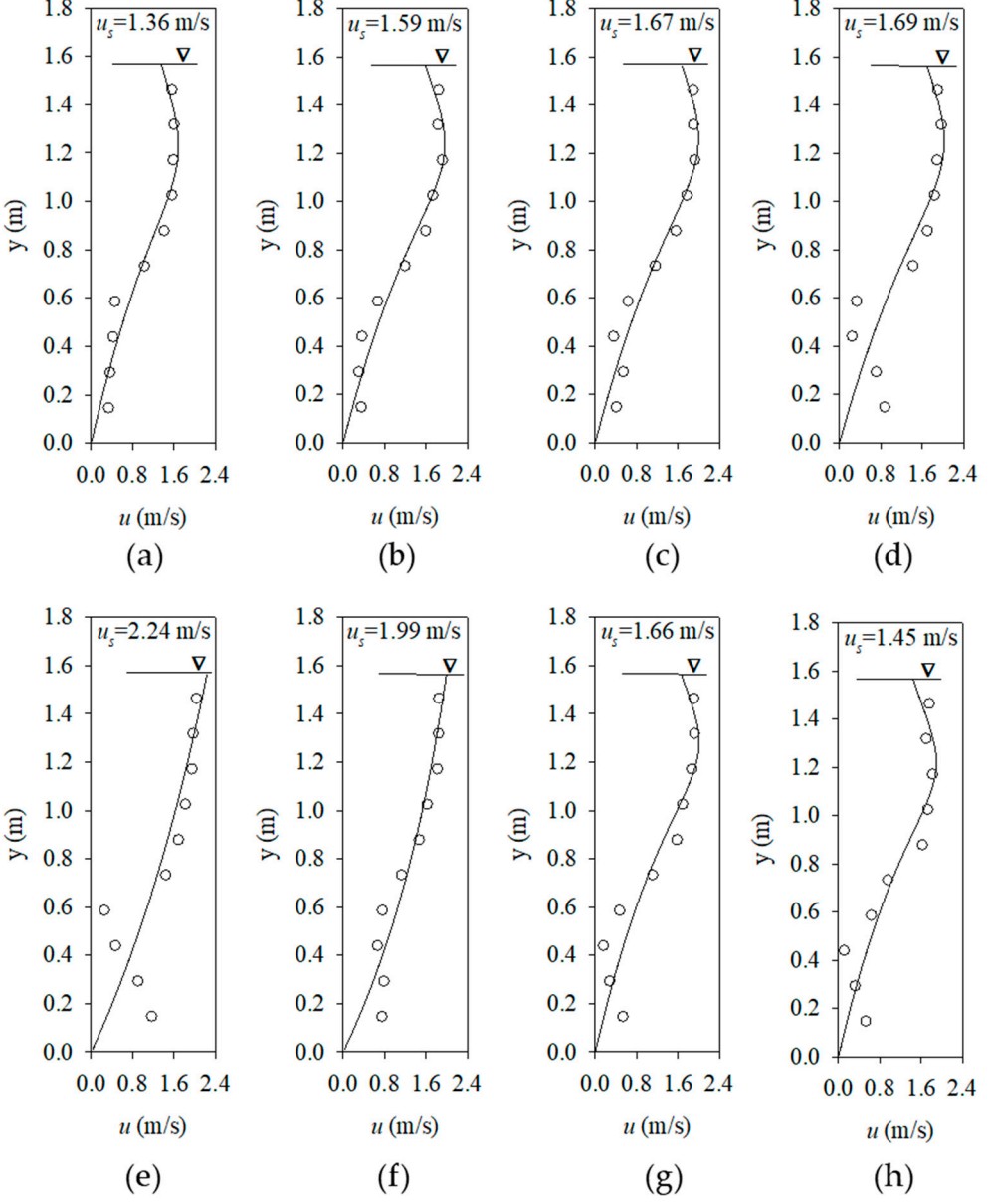

Note:
u: velocity; u_S: surface velocity

**Figure 7.** Subsectional velocity profile of the Run 4 in the Longen Channel. (**a**) vertical I; (**b**) vertical II; (**c**) vertical III; (**d**) vertical IV; (**e**) vertical V; (**f**) vertical VI; (**g**) vertical VII; (**h**) vertical VIII of Figure 6.

## 4. Results and Discussion

The discharge measurement in the natural channel at the Guanyin Bridge was taken by the conventional two-point method for the water depth and the mean vertical velocity. Meanwhile, owing to the limitations caused by the channel width, the discharge measurement at the Longen Channel was taken from the vertical velocity profile for the mean velocity. The mid-section method was then applied to both the natural and the artificial channels. The mean cross-sectional velocity can be observed by the discharge divided by the cross-sectional area. Figure 7 shows the vertical velocity profile of the Longen Channel on March 8, 2019 when the discharge was 3.80 m$^3$/s. The flow pattern of the Longen Channel was quite similar to a large-scale hydraulic flume in a laboratory. In Figure 7, it is obvious that the flow patterns of Longen Channel were very different from those of natural rivers. Most of the maximum velocities on the verticals occurred at a depth of about 1/4 water depth from the water surface, while the surface velocity was relatively small. It also shows that the maximum velocities of the verticals excluding verticals (e) and (f) did not occur on the water surface. Experimental studies have been shown from considerations of momentum transfer that the velocity in an open channel should decrease toward the channel bed. In a very wide channel the velocity decreases toward the bed and walls, and theoretically the maximum occurs at the water surface. The Longen Channel is a small artificial flume; therefore, depression of the maximum velocities below the water surface was observed. The flow pattern cannot be described by a logarithmic distribution. The circle in Figure 7 is the actual velocity measurement on each vertical, and the line is the velocity distributions based on (11) indicating that vertical maximum velocity does not always occur on the water surface. It also shows that the velocity profile data of the Longen Channel is difficult to describe using conventional velocity distribution theories, such as logarithm velocity distribution. However, (11) can simulate velocity profiles effectively, regardless of whether the maximal velocity occurs on or below the water surface. Therefore, the surface velocities on the verticals could also obtained precisely by using (11). In addition, using the nonlinear regression method, $M$, $h$ and $u_{max}$ can also be obtained from (11) with the vertical velocity and water depth. Thus, the mean vertical velocity ($\overline{u}_i$ $in$ [1]) on each vertical can be estimated. Therefore (11) can be used to accurately estimating the mean velocity of the vertical for obtaining reliable discharge.

Figure 8 shows the relationship of the mean surface velocity and mean velocity of Nankang River at Guanyin Bridge and at Longen Channel. The surface velocity of the natural river was directly measured by SVR, while surface velocities of artificial channels were estimated by the probabilistic velocity distribution equation. All the points, including those in the left and right channels, distribute closely on the two sides of the regression. The relationships of the mean surface velocity and the mean velocity of the cross section in the left and right channels in Figure 8a do not show much difference.

When all the points were combined, all the points fall on the periphery of the regression line, and when the mean surface velocity increases, the data tends to approach the regression line. This means that as the water depth increases, the width of the cross-section increases even more making the shape of the cross-section approach the shape of a rectangle. Hence, the relationship between the mean surface velocity and the mean velocity was stable. Figure 8b shows an artificial rectangular channel with all the points falling near the regression line. This means that the relationship between $\overline{u}_s$ and $\overline{u}_{obs}$ of the artificial rectangular channel is very stable. It reveals that the mean cross-sectional velocity and mean surface velocity in both the natural and the artificial channels have correlation coefficients of 0.92 and 0.99, respectively. As the artificial channel is confined with the concrete bed and walls, the velocity is less affected by natural factors; and thus, it performs better than the natural channel. Figure 8 demonstrates that the relationship between the mean surface velocity and the mean velocity of natural rivers and artificial channels is quite stable, forming a linear relationship through the origin.

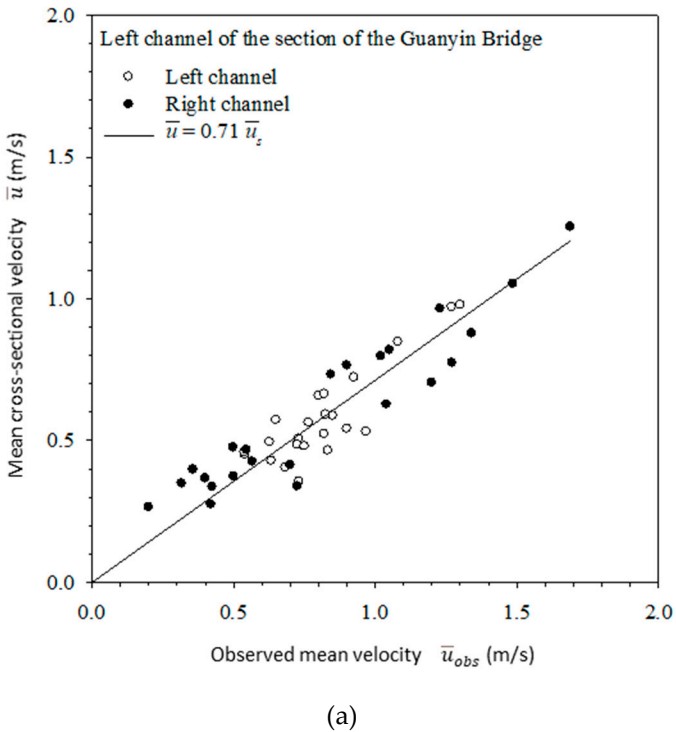

(a)

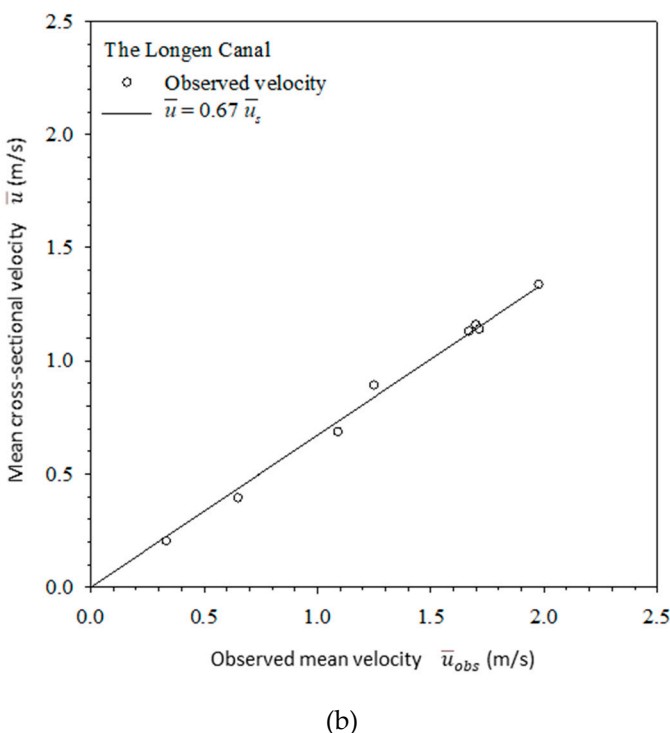

(b)

**Figure 8.** Relation between mean velocities of cross section and surface. (**a**) the Nankang River at the Guanyin Bridge; (**b**) the Longen Channel.

Nangang Creek is a mountain river at Guanyin Bridge, with steep slopes and continuous erosion and sediment deposition. When a flood occurs, the river course will initially be deepened, although the river course will be silted back in as the water recedes. However, the data from 2018 show that the water stage did not change much during the study; nevertheless, the flow and cross-sectional area of the river have changed greatly over time due to erosion and deposition along the river. Thus, it is impossible to estimate the water

cross-sectional area based on the water stage. In order to quickly estimate the cross-sectional area, the water depths of the left and right channels at 30 m and 91 m from the initial point were used to establish the relationship between water depth and cross-sectional area (Figure 9). The points (dots and open circles) in the figure represent observed data in the right and left channels, respectively, while the lines represent the depth-cross-sectional area rating curves obtained by nonlinear regression. The nonlinear regression method was applied to relate the water depth and the cross-sectional area with the coefficients given from the observations. The points did not deviate too far from the regression lines. The correlation coefficients of the left and right channels were 0.94 and 0.90, respectively, suggesting that the regression equations are quite useful for making estimations of cross-sectional area by using depths.

As the shape of the artificial channel is rectangular, the cross-sectional area can be obtained easily from the water depth and the width of the channel. Therefore, the cross-sectional area of the Longen Canal can be quickly and accurately estimated by the water depth of each run and the width of the channel.

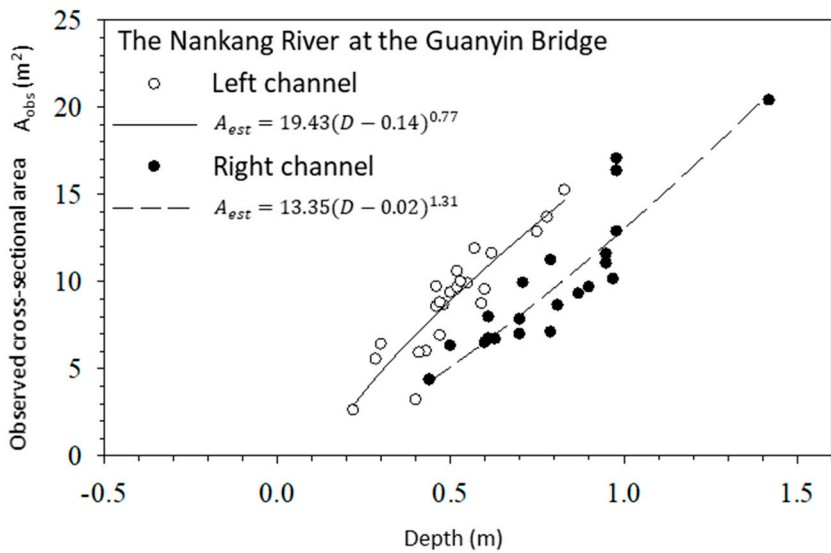

**Figure 9.** Relation between gage height and cross section area at the Guanyin Bridge.

After the two relationships: (1) between mean cross section and mean surface velocity, and (2) between water depth and cross-sectional area were established, the estimated discharge could be quickly obtained with mean surface velocity and water depth. The estimated discharge with the mean surface velocity and water depth were:

$$Q_{est} = 13.77\overline{u}_s(D - 1.35)^{0.78} \text{ for the left channel of the Nangang River} \qquad (15)$$

$$Q_{est} = 9.48\overline{u}_s(D - 0.02)^{1.31} \text{ for the right channel of the Nangang River} \qquad (16)$$

$$Q_{est} = 1.74\overline{u}_s D \text{ for the Longen Canal} \qquad (17)$$

Figure 10 illustrates the accuracy of the discharge measurement using the mean surface velocity. The $x$- and $y$-axes represent the discharge measured by the mean surface velocity and the conventional method, respectively. This figure also shows that the natural channel cross-section was close to a rectangle; however, the estimation of the cross-sectional area of the natural channel was not as accurate as that of the artificial channel. As a result, the measurement of the artificial channel was better than that of the natural channel, but all points fall on the 45-degree agreement line, which demonstrates that the streamflow measured by both the conventional and the proposed methods were close to each other with less than 1% error on average.

A strong correlation ($R^2 > 0.97$) between the above two methods demonstrates the accuracy and reliability of using the mean surface velocity method for discharge measurement in both the natural rivers and artificial channels. Therefore, the authors concluded the measurement of river discharge can be obtained promptly and accurately using the proposed approach, which only requires one to measure the surface velocity to obtain the mean surface velocity, and estimate the cross-sectional area based on the water depth (or water stage).

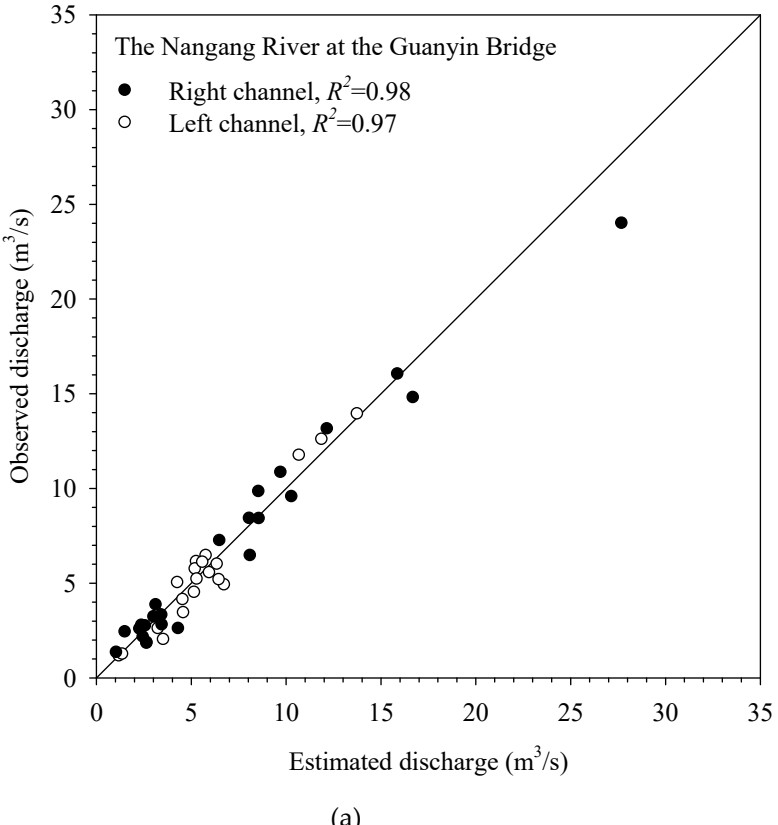

(a)

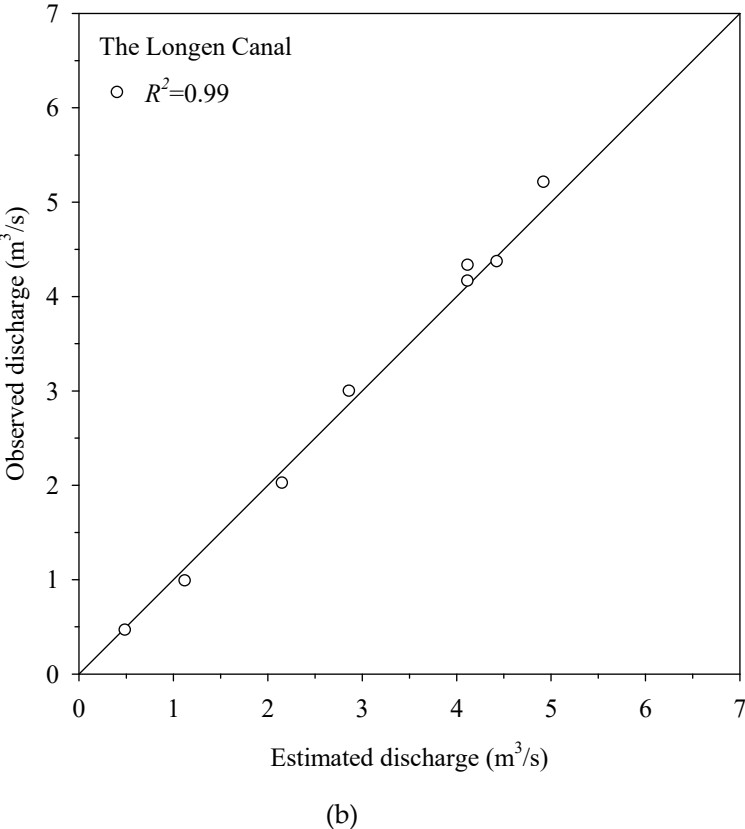

(b)

**Figure 10.** Accuracy of discharge measurement by surface mean velocity; (**a**) the Nankang River at the Guanyin Bridge; (**b**) the Longen Channel.

## 5. Conclusions

The conventional methods for discharge measurement consume more labor and time than the proposed approach. Particularly, it is more risky when taking discharge measurement during the high water. The proposed approach applies the relationship between the mean cross-sectional velocity and the mean surface velocity as a constant in each cross section if the cross section does not change too much. Once the constant of the relationship is found, the discharge can be easily estimated based on the constant, the mean surface velocity, and the cross-sectional area. The mean surface velocity can be obtained through the SVR or other means; and the cross-sectional area can be estimated by water depth (or stage) or other means. Nowadays, the surface velocity radar (SVR) has become a popular instrument, which makes the surface velocity measurement easy, reliable and accurate.

The data collection from a natural stream and an artificial channel was used to demonstrate the proposed approach. Two case studies were conducted at two locations with many session runs at different times with low to intermediate flows. Both conventional and the proposed methods were conducted for flow measurements. The results show the feasibility of measuring streamflow with an accuracy of less than 1% difference on average comparing the proposed and the conventional methods. Moreover, the strong correlation coefficient (>0.97) of the observed discharge and the estimated discharge suggests the reliability of the proposed approach for streamflow measurement in both the natural and artificial channels. In previous studies, the conventional methods required extra coefficients (velocity profile or surface velocity coefficient) of each subsection for determining the mean velocity in each subsection, involving more time and labor cost. In this study, we proposed that the mean surface velocity to estimate streamflow can provide an opportunity to improve the conventional methods in streamflow measurement and maintain the accuracy of discharge measurement.

This approach can substantially reduce the uncertainty involved in determining surface velocity coefficients that are employed in conventional methods, while maintaining the accuracy of discharge measurement. Compared with the conventional methods, the proposed approach saves more labor and time cost. As it is a non-contact method, it can also reduce the risk to human life and measuring instruments when taking measurements in the natural environment. The proposed approach can be applied in both natural and artificial channels for flow measurement. Based on the measurement sessions implemented in this study, we conclude that the proposed approach can provide reliable and accurate streamflow measurement from low to intermediate flows. The constancy of the relationship between the mean cross-sectional velocity and the mean surface velocity suggests that this approach might also be applied for high water conditions, which would need further tests in the future.

**Author Contributions:** All authors have read and agreed to the published version of the manuscript. Conceptualization, Y.-C.C.; Data curation, E.O.Z., Formal analysis, Y.-C.H.; Funding acquisition, Y.-C.C.; Investigation, Y.-C.H. and E.O.Z.; Methodology, Y.-C.C.; Supervision, Y.-C.C.; Validation, Y.-C.H.; Writing – original draft, Y.-C.H. All authors have read and agreed to the published version of the manuscript.

**Funding:** This article was based on work supported by the Ministry of Science and Technology, Taiwan (Grant no. MOST 110-2221-E-027-021-).

**Institutional Review Board Statement:** Not applicable.

**Informed Consent Statement:** Not applicable.

**Data Availability Statement:** The data presented in this study are available on request from the corresponding author.

**Acknowledgments:** The authors would like to acknowledge the Third River Management Office and the Northern Region Water Resources Office of the Water Resources Agency, Taiwan for their kind assistance in measurement sessions.

**Conflicts of Interest:** The authors declare no conflicts of interest.

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
