# Peer review of "Streamflow Measurement Using Mean Surface Velocity"

_water, doi:10.3390/w14152370_

Round 1

Reviewer 1 Report

Please see the attached comments

Reviewer 2 Report

The manuscript is dedicated to the method which can be applied for estimating the streamflow of natural streams and artificial channels. The authors propose to use the mean surface velocity to estimate streamflow that can provide an opportunity to improve the conventional methods in streamflow measurement and maintain the accuracy of discharge measurement. They have demonstrated that proposed approach is capable to provide reliable and accurate streamflow measurement. The results obtained are new, interesting, and valuable for the field. The results are clear, and their analysis is also presented well. The paper is written well, but it should be reorganized to be in accordance with the journal requirements to the structure of the article. So, my opinion is that the paper needs at least minor revision. I suggest the authors make the following corrections before its publication:

1. Please, prepare names and affiliation lines in the paper in accordance with the journal template:

“Firstname Lastname 1, Firstname Lastname 2 and Firstname Lastname 2,*

1 Affiliation 1; e-mail@e-mail.com

2 Affiliation 2; e-mail@e-mail.com

* Correspondence: e-mail@e-mail.com; Tel.: (optional; include country code; if there are multiple corresponding authors, add author initials)”.

If it is still not understandable, please, look at some latest published papers in the journal, e.g., https://www.mdpi.com/2073-4441/14/9/1409.

2. The paper should have the following structure (see the journal template): Introduction, Materials and Methods, Results, Discussion, and Conclusions. Please, reorganize your paper so as to provide section Materials and Method instead of current section 2. Methodology.

3. Introduction.

Please, separate references ‘[3-14]’ for appropriate groups of references and provide their short characterization. To do so, please, group these references properly providing their main features.

4. Figure 1 caption.

Please, provide description for all the parameters presented by variables in Figure 1: b1, b2, b3,…, dn-1.

5. Please, outline in Conclusion other possible applications of the method and algorithm proposed, and future research perspective.

6. Please, rewrite Author Contributions in accordance with the journal requirements:

Author Contributions: For research articles with several authors, a short paragraph specifying their individual contributions must be provided. The following statements should be used “Conceptualization, X.X. and Y.Y.; methodology, X.X.; software, X.X.; validation, X.X., Y.Y. and Z.Z.; formal analysis, X.X.; investigation, X.X.; resources, X.X.; data curation, X.X.; writing—original draft preparation, X.X.; writing—review and editing, X.X.; visualization, X.X.; supervision, X.X.; project administration, X.X.; funding acquisition, Y.Y. All authors have read and agreed to the published version of the manuscript.” Please turn to the CRediT taxonomy for the term explanation. Authorship must be limited to those who have contributed substantially to the work reported.”

7. The “Conflicts of Interest” paragraph is missing. Please, provide it.

8. Please, prepare references exactly in accordance with the journal template, provide missing DOIs where possible and abbreviate journal names:

1. Author 1, A.B.; Author 2, C.D. Title of the article. Abbreviated Journal Name Year, Volume, page range.

2. Author 1, A.; Author 2, B. Title of the chapter. In Book Title, 2nd ed.; Editor 1, A., Editor 2, B., Eds.; Publisher: Publisher Location, Country, 2007; Volume 3, pp. 154–196.

3. Author 1, A.; Author 2, B. Book Title, 3rd ed.; Publisher: Publisher Location, Country, 2008; pp. 154–196.

4. Author 1, A.B.; Author 2, C. Title of Unpublished Work. Abbreviated Journal Name year, phrase indicating stage of publication (submitted; accepted; in press).

5. Author 1, A.B. (University, City, State, Country); Author 2, C. (Institute, City, State, Country). Personal communication, 2012.

6. Author 1, A.B.; Author 2, C.D.; Author 3, E.F. Title of Presentation. In Proceedings of the Name of the Conference, Location of Conference, Country, Date of Conference (Day Month Year).

7. Author 1, A.B. Title of Thesis. Level of Thesis, Degree-Granting University, Location of University, Date of Completion.

8. Title of Site. Available online: URL (accessed on Day Month Year).”

If it is still not understandable, please, look at some latest published papers in the journal, e.g., https://www.mdpi.com/2073-4441/14/9/1409.

So, the paper needs minor revision.

Reviewer 3 Report

The article is correctly written at the right level. The presented method is interesting and important due to the conduct of scientific research. Only the introduction should be improved and the citations changed. Below is a detailed note on what I mean.

Line 45-46: "...replace mechanical current meters for velocity measurement [3-14] Ground penetration radar..." - this citation is a joke. Most of these articles are not re-cited!! Please leave a maximum of 3 items at this point and delete the rest.

Round 2

Reviewer 1 Report

Few suggestions were followed.  Some basic rules in writing scientific manuscript were not followed, such as

·        The objectives and the conclusions should agree with the results.

·        Figures and tables should be understandable without reading the text.

I was disappointed that the suggestions were not implemented.  It is the reason for my recommendation of  major revisions.  The changes are crucial for acceptance for publication.  Detailed comments are in the attached file

Round 3

Reviewer 1 Report

Dear Authors

The manuscript is more realistic.   It is not as much over the top as the last version of the manuscript.  The authors should check their spelling before submitting and still, not all symbols and abbreviations are not explained in the figure titles.  Figure 6 should have a better explanation of what the red lines and the roman numerals mean. 

Author Response

This manuscript is a resubmission of an earlier submission. The following is a list of the peer review reports and author responses from that submission.

Round 1

Reviewer 1 Report

This is a resubmitted manuscript I reviewed before. Overall, the authors have addressed all my comments. It should be accepted for publication now.

Reviewer 2 Report

Dear Authors

Some more work is required before it can be published. Please see the attachment

Reviewer 3 Report

The paper presents a method for the determination of the discharge using surface velocity measurements. I have a few comments/questions:

  1. there is a typo in line 15, "natural steam" instead of "natural stream";
  2. lines 81-87 are a bit messy, looks like some equations appear before being introduced and/or are not introduced. Please fix it;
  3. please check the axis and labels of figure 4. What depth are you referring to? Since the water surface is not at 0, that's clearly not water depth. Is it maybe the distance from the bottom of the bridge? If that's the case, please change the label accordingly. Similar issue with the horizontal label: "width" would be appropriate if you were drawing a scatter plot of depth and width of different objects, but not to indicate a coordinate, which is a distance from a reference point. Is there a reason why it goes from 86 to 109 rather than from 0 to 23?
  4. In figure 6 you have placed ticks for the vertical direction with a non-dimensional progressive coordinate in the centerline of the measuring points, while in the horizontal direction the ticks are aligned with the measuring points, with constant length values. Can you please draw again the figures in a coherent way? What's the distance of sections I and VIII from the wall? Could you please add it to the description in section 3?  
  5. in line 235 "because" seems to be completely out of context. Don't you think that something like "when" or "as" would be more appropriate?
  6. "This means that as the water depth increases, the shape of the cross-section approaches the shape of a rectangle": that's interesting, please discuss this aspect more in detail.
  7. lines 233-241: this part is basically repeating in a loop that the relation between u_s and u_obs is very stable. Please be more concise.
  8. why the part from line 293 to line 301 is in the conclusions? It's definitely misplaced but could be suitable for an introduction perhaps. Could you please delete it or move it in the introduction?
  9. "an accuracy of less than 1% error on average" doesn't sound like a sentence of a scientific paper. First, can you please define which metric are you using to measure accuracy? Otherwise, that 1% is not very meaningful.
    Second, the syntax is awful. Are you referring to the accuracy or to the error? Please be precise.
  10. "the strong correlation coefficient of >0.98": saying that a coefficient is strong sounds weird. You could rather say that there is a strong correlation, or that a coefficient has a high value. However, that adds nothing to the discussion if you already have said that the accuracy is high.
  11. "Using mean surface velocity to estimate streamflow can provide an opportunity to improve the use of surface velocity to measure streamflow": this doesn't seem to match what is shown in the discussion of the results. It is not clear how the proposed method performs in comparison with the method proposed by reference 25

In general, after reading the paper it's still not clear what's the advantage of this method in comparison with the one presented in reference 25. Looks like it's still necessary to get a few velocity samples on the surface velocity to measure the discharge, but looks like the presented method is reliable only with rectangular sections, right?

Results and discussions and Conclusions appear vague and verbose, please revise it focusing on how the proposed method performs in comparison to other methods based on surface velocity, perhaps taking as a reference the mid-section method.